# Method for Calculating the Required Number of Transport Vehicles Supplying Aviation Fuel to Aircraft during Combat Tasks

**Jarosław Ziółkowski** [1], **Józef Żurek** [2], **Jerzy Małachowski** [1,*], **Mateusz Oszczypała** [1] and **Joanna Szkutnik-Rogoż** [1]

1. Faculty of Mechanical Engineering, Military University of Technology, 00-908 Warsaw, Poland; jaroslaw.ziolkowski@wat.edu.pl (J.Z.); mateusz.oszczypala@wat.edu.pl (M.O.); joanna.szkutnik@wat.edu.pl (J.S.-R.)
2. Department of IT Logistics Support, Air Force Institute of Technology, 01-494 Warsaw, Poland; jozef.zurek@itwl.pl
* Correspondence: jerzy.malachowski@wat.edu.pl; Tel.: +48-261-839-140

**Abstract:** During aircraft flights, combat readiness and the supply system affecting it are essential issues. The basic items of supply during the implementation of tasks are combat assets and aviation fuel. Effective management of the flow of required products, as well as the reliability of vehicles and the availability of crews contribute to the quality of task performance. The components that make up this quality in military operations are measured by readiness. In real-life operations, the number of vehicles supplying aircraft with aviation fuel is determined for safety and reliability with an surplus related to the number of flight support facilities. This paper develops a method for determining the minimum number of vehicles required to supply aircraft (*sp*) with aviation fuels. The developed method was verified by a numerical example illustrating its application in practice. Additionally, a detailed analysis of its application was carried out in relation to potentially 50 possible scenarios of combat task execution, with a number of assumptions fulfilled. Based on the performed calculations, it was concluded that the number of vehicles required for *sp* fuel supply depends on several factors: the number of aircraft, the characteristics of air tasks (flight length and frequency of departures), as well as the time of clean *sp* refuelling and the duration of the vehicle-tanker refuelling cycle.

**Keywords:** vehicles; aircraft; delivery; optimization

## 1. Introduction

The issues related to the optimization of a process, phenomenon, system, or object constitute a wide spectrum of interest and have a significant publishing potential. They naturally position the possibilities of improving the areas requiring improvement. The issues of optimization are located in the area of linear programming [1–3], where the end result is the achievement of the assumed optimization goal. In practice, the assumed function of the objective may therefore be differently defined, for example, to strive to achieve a certain extreme in the form of minimizing inputs or maximizing profits. The collection of publications that define it as the minimum value of the objective function includes, for example, the works of [4–6]. In other studies, the authors proposed to shorten the time [7] of delivery of perishables [8], reduction in empty cargo runs [9], optimization of the departure schedule while taking into account arrival intersections [10], selection of an optimal investment variant [11], cost minimization of transport [12] supply chain [13], product lifecycle [14], the choice of a logistics operator [15], estimation of aircraft fuel consumption [16], or the number of vehicles supplying aviation fuel to aircraft during flights [17]. The set of publications with the objective function defined as the maximum is slightly more modest. It includes articles on how to effectively increase the impact in

a linear model of threshold propagation in linear programming [18], solving the internal problem of maximization using the double gradient method [19], increasing the revenues obtained in a stable power grid [20], or simply increasing efficiency [21].

A separate collection consists of studies in which the authors deal with the issues of multicriteria methods of analysis and evaluation. They apply to a wide range of activities, including but not limited to:

- The field of artificial intelligence; for example, in the form of a multipurpose firework algorithm based on the Pareto domination method to solve problems within multimodal transport networks [22];
- Three-dimensional: distance transformation to optimize restricted mountain railway routing [23], and the problem of routing vehicles with limited capacity taking time gaps into account [24];
- Double optimization for lane reservation with remaining capacity and budget constraints [25];
- Multipurpose transport problem with Pareto optimality criteria (matrix maxima method) [26];
- Issues related to the determination of the location of the warehouse within the logistics network [27];
- Multipurpose optimization of custom bus routes in real time based on a two-step approach [28];
- The problem of expanding the capacity of renewable sources under conditions of uncertainty [29];
- Mapping routes in open locations in many depots with a heterogeneous fixed fleet [30].

The narrowest area of the subject consists of a few studies concerning improvements in the area of security and reliability in military systems [31,32]. For example, Homsi [33] developed two general approximation algorithms (heuristic and metaheuristic) to refurnish a unit (the so-called multiple-backpack problem) with many different goods available in different locations. On the other hand, Bacanin et al. [34] used the so-called whale optimization algorithm for locating wireless sensor networks.

In operational practice, it is not always possible to improve a given area in a satisfactory manner by achieving the assumed target function. In such a case, the obtained result is interpreted as the initial or initial base solution [35]; i.e., one that meets the assumed limiting conditions, but is not optimal.

As a rule, publications in the area of linear programming present new methods [36,37], techniques [38], algorithms [39], or improvements [40] to existing methods. On the other hand, solutions that could be defined as innovative are less common [41].

In this study, the authors dealt with the original and unique issue related to the problem of determining the minimum number of vehicles supplying aviation fuel to aircraft performing combat tasks. The Su-22 combat fighter aircraft (manufactured in the USSR) is still used by the air force units of the Republic of Poland (RP). During the execution of combat tasks, after departure, the readiness for the next flight is regained each time, the essential element of which is refueling. The refueling process is carried out using two types of tanker vehicles of different tank capacities, 4500.0 ($dm^3$) and 7500.0 ($dm^3$), respectively. According to the operational practice, the number of vehicles of a certain type needed to secure aviation fuel is determined arbitrarily each time with a set operational (equipment) surplus. This is due to safety reasons, and is dictated by the reliability of the fuel supply system, ensuring the performance of planned combat tasks. The number of fuel delivery vehicles depends on a number of factors, including the number and type of aircraft (*sp*) involved in the flights, the capacity of the tank(s) of the main (or all) aircraft, and the emptying rate. The adopted organization of flights is also important, including both the length of individual flights and the frequency of departures individually for each aircraft. The proposed method in this publication enables the determination of the necessary (minimum) number of vehicles needed to secure necessary aviation fuel for the aircraft. It is dedicated especially to aviation tasks carried out in emergency situations;

for example, during war operations, when random losses of military equipment (*ME*), both in terms of primary and supporting equipment often being of essential operational importance, are characteristic. The developed method is universal and can be applied to any type/kind of aircraft and any combat flight scenario.

In order to present the solution to this original problem, in which the authors have offered their contribution to the solution of a unique issue, the following publication layout has been proposed. Section 1 presents a literature review of linear programming, which is a method used to achieve the best (optimal) solution in the decision-making process. In Section 2, all assumptions and development of a mathematical model developed on the basis of an analysis of the real process of aviation fuel supply of Su-22 aircraft performing combat tasks are presented. The proposed model is an original scientific achievement of the authors of this publication. Section 3 illustrates the application of the developed model on a numerical example related to the real scheduled flight table (real case) reflecting the combat tasks performed by the tactical aviation base of the Polish Armed Forces. Section 4 illustrates the extension and generalization of the developed mathematical model by carrying out calculations for 50 potential (possible) action scenarios. The obtained results were a confirmation of correctness and at the same time universality of the proposed model. Finally, in Section 5, the obtained solutions indicating shortcomings and factors affecting the obtained results are discussed and concluded.

## 2. Assumptions and Development of the Mathematical Model

The following assumptions were adopted for the development of the mathematical model enabling the calculation of the necessary number of vehicles supplying fuel to aircraft:

- The number of aircraft performing aviation tasks is a random step variable;
- The emptying factor of the main tank of the aircraft is a random step variable;
- Vehicles break down during the execution of tasks at random moments;
- The vehicle replacement time (conversion to a technically efficient one) is strictly determined;
- The duration of the flights is fixed;
- Execution of aviation tasks performed by aircraft (in accordance with a scheduled flight table).

The following markings were adopted for the development of the model:

- Number of aircraft, $N_{SP}$;
- The capacity of the main aircraft tank(s), $V_{zbsp}$;
- The emptying factor of the aircraft fuel tank, $K_{zu}$;
- Number of vehicles supplying aviation fuel to aircraft, $N_P$;
- Capacity of the vehicle supplying aircraft with aviation fuel, $V_P$;
- Flight time, $T_0$.

The fuel balance equation for one flight ($V_{elt1}$) of the aircraft in accordance with the assumed flight duration $T_0$ can be determined according to the formula:

$$V_{elt1} = \sum_{l=1}^{N_{elt}} K_{zu} \cdot V_{zbsp} \tag{1}$$

where:

$N_{elt}$—number of aircraft takeoffs;
$l$—the $l$-th flight of the aircraft.

The equation of the fuel balance for the maximum number of aircraft flights of a tactical fighter squadron can be written as:

$$V_{eltmax} = \sum_{k}^{N_{SP}} \sum_{l}^{N_{elt}} (K_{zu}^{lk}) \cdot V_{zbsp} \tag{2}$$

where:

$N_{SP}$—number of aircraft of a tactical fighter squadron performing aviation tasks;
$K_{zu}^{lk}$—fuel tank emptying factor for the *l*-th flight of the *k*-th aircraft;
$N_{elt} \in \{1, \dots, 8\}$ takeoff of the *k*-th aircraft of the tactical fighter squadron.

Dependency (2) takes into account the assumption that flights were performed by all aircraft. If this is not the case, then the *l*-th takeoff of the *k*-th aircraft should be zero $\left(K_{zu}^{lk} = 0\right)$.

The refueling of the Su-22 aircraft was considered in two scenarios; i.e.:

1. Assuming zero waiting time ($t_{ocz} = 0$);

$$t_t = t_m + K_{zu} \cdot t_{et} \tag{3}$$

where:

$t_t$—aircraft refueling time;
$t_m$—handling time (related to the vehicle's travel time to the aircraft, the time needed to connect the filling nozzle for refueling, etc.);
$K_{zu}$—the tank emptying factor of the aircraft;
$t_{et}$—time that it takes the vehicle to refuel an empty aircraft tank.

2. Assuming the maximum frequency of takeoffs (e.g., every 40 min) for which $t_{ocz} \in (40 - t_t)$.

The maximum amount of required fuel is obtained when flights are carried out by a tactical fighter squadron equipped with *n* = 16 aircraft for the longest flight duration of 50 min with the assumed frequency of takeoffs every 40 min. Taking into account the flight duration time horizon of 8 h, each facility will execute a maximum of 5 takeoffs in this interval, in accordance with (4):

$$V_{eltmax} = \sum_{k=1}^{16} \sum_{l=1}^{5} K_{zu} \cdot V_{zbsp} = 307{,}100 \, [\text{dm}^3] \tag{4}$$

Figure 1 shows the curves illustrating fuel consumption by a tactical fighter squadron depending on the length of a single flight. It was assumed that 7 to 16 aircraft participated in the flights, which was reflected in the coefficient of efficiency $W_{efelt} \in \{0.43; \dots; 1\}$ of the utilization of aircraft of the tactical fighter squadron (the efficiency of the use of the aircraft fleet is understood as the quotient of the number of aircraft participating in the flights to the total number of aircraft in the tactical fighter squadron).

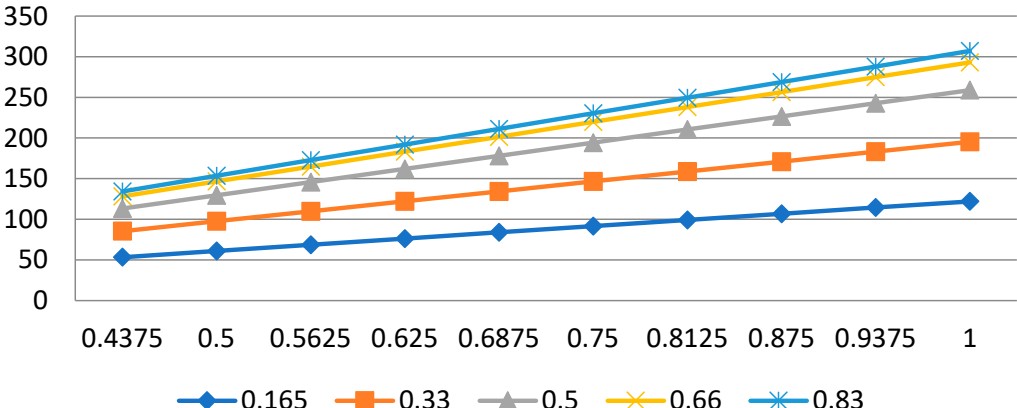

**Figure 1.** Fuel consumption (m$^3$) by the tactical fighter squadron depending on factor $K_{zu} = \{0.165; 0.33; 0.5; 0.66; 0.83\}$ and the efficiency factor $W_{efelt} \in \{0.43; \dots; 1\}$ of the number of aircraft used.

An analysis of Figure 1 shows that fuel consumption is linear (or almost linear). The highest occurs during the longest flights, up to 50 min, for which the fuel emptying factor

of the aircraft is $K_{zu} = 0.83$ Having determined the amount of fuel required, in the next step, we should calculate the number of vehicles with 7500.0 $(dm^3)$ capacity needed to transport the required amount of fuel (Figure 2).

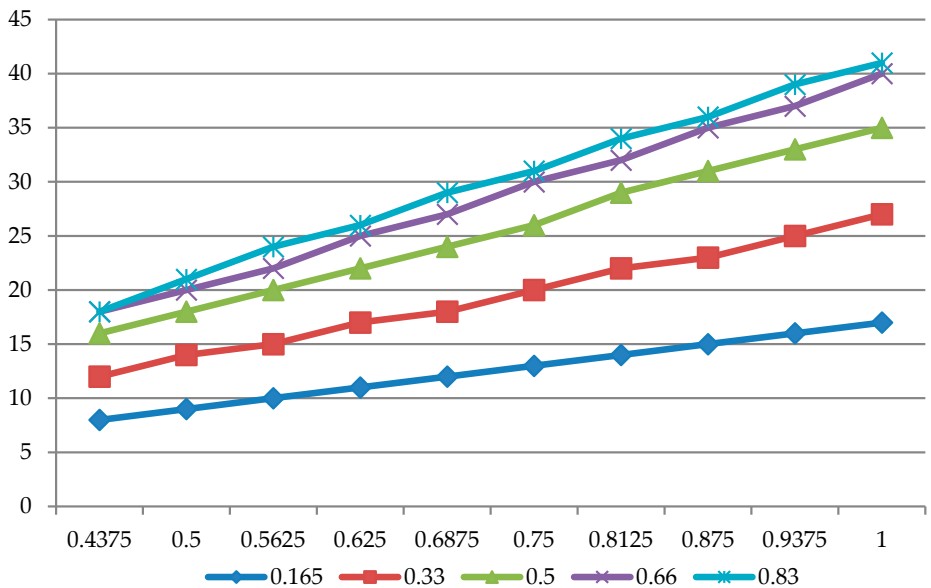

**Figure 2.** Number of vehicles with the capacity of 7500.0 $(dm^3)$ volumetrically needed to transport fuel consumed by a tactical aviation squadron, depending on the flight distance and the efficiency coefficient $W_{efelt} \in \{0.43; \ldots ; 1\}$ of the number of aircraft used.

The curves in Figure 2 illustrate a significant variation in the number of vehicles required to carry the necessary amount of fuel. This depends on the efficiency ($W_{efelt}$) of the utilization of the aircraft fleet and the duration of a single flight. Detailed data are summarized in Tables 1 and 2, respectively. They show the minimum number of vehicles necessary to ensure continuity of fuel supplies. Required vehicles with the capacity of 7500.0 $(dm^3)$ range from $8 \div 41$, and required vehicles with the capacity of 4500.0 $(dm^3)$ range from $12 \div 69$.

The results summarized in Tables 1 and 2 show the ranges of variation in the number of vehicles with specific capacities, respectively (7500.0 $(dm^3)$ for Table 1 and 4500.0 $(dm^3)$ for Table 2), needed by volume to deliver the necessary amount of fuel (per flight) to secure flights. The $W_{efelt}$ coefficient of aircraft utilization efficiency, ranging from 43.75% to 100%, means that 7 to 16 aircraft on the equipment of the tactical aviation squadron are actively involved in the execution of combat tasks. As can be seen from the above tables, the needs for fuel supply directly translate into the number of vehicles of a certain capacity according to the principle of the smaller the capacity, the more vehicles should be secured to carry the required amount of fuel in one trip.

**Table 1.** Number of vehicles with a capacity of 7500.0 $(dm^3)$ needed to transport the required amount of fuel in one run, depending on the flight duration $K_{zu} \in \{0.165; 0.33; 0.5; 0.66; 0.83\}$ and the $W_{efelt}$ efficiency factor for the utilization of the aircraft fleet.

| $W_{efelt}$ | 0.4375 | 0.5 | 0.5625 | 0.625 | 0.6875 | 0.75 | 0.8125 | 0.875 | 0.9375 | 1 |
|---|---|---|---|---|---|---|---|---|---|---|
| $K_{zu} = 0.165$ | 8 | 9 | 10 | 11 | 12 | 13 | 14 | 15 | 16 | 17 |
| $K_{zu} = 0.33$ | 12 | 14 | 15 | 17 | 18 | 20 | 22 | 23 | 25 | 27 |
| $K_{zu} = 0.5$ | 16 | 18 | 20 | 22 | 24 | 26 | 29 | 31 | 33 | 35 |
| $K_{zu} = 0.66$ | 18 | 20 | 22 | 25 | 27 | 30 | 32 | 35 | 37 | 40 |
| $K_{zu} = 0.83$ | 18 | 21 | 24 | 26 | 29 | 31 | 34 | 36 | 39 | 41 |

**Table 2.** Number of vehicles with a capacity of 4500 (dm$^3$) needed to transport the required amount of fuel in one run, depending on the flight duration $K_{zu} \in \{0.165; 0.33; 0.5; 0.66; 0.83\}$ and the $W_{efelt}$ efficiency factor for the utilization of the aircraft fleet.

| $W_{efelt}$ | 0.4375 | 0.5 | 0.5625 | 0.625 | 0.6875 | 0.75 | 0.8125 | 0.875 | 0.9375 | 1 |
|---|---|---|---|---|---|---|---|---|---|---|
| $K_{zu}$ = 0.165 | 12 | 14 | 16 | 17 | 19 | 21 | 23 | 24 | 26 | 28 |
| $K_{zu}$ = 0.33 | 19 | 22 | 25 | 28 | 30 | 33 | 36 | 38 | 41 | 44 |
| $K_{zu}$ = 0.5 | 26 | 29 | 33 | 36 | 40 | 44 | 47 | 51 | 54 | 58 |
| $K_{zu}$ = 0.66 | 29 | 33 | 37 | 41 | 45 | 49 | 53 | 57 | 62 | 66 |
| $K_{zu}$ = 0.83 | 30 | 35 | 39 | 43 | 47 | 52 | 56 | 60 | 64 | 69 |

The balance equations of capacity and working time for the vehicle are described by the relations (5) and (6), respectively:

- Capacity balance:

$$N_{PV} \cdot V_P \geq V_{eltmax} \quad \text{hence}: \ N_{PV} \geq \frac{V_{eltmax}}{V_P} \tag{5}$$

where:

$N_{PV}$—the number of vehicles required to carry the fuel consumed by the tactical fighter squadron in the operation of flights;

$V_P$—capacity of the aviation fuel delivery vehicle;

$V_{eltmax}$—maximum fuel consumption by the tactical fighter squadron during flights.

- Work time balance: it was assumed that the time needed for the vehicle refueling cycle $t_{up}$ must meet the condition:

$$t_{up} \leq t_{nPV} + t_m + t_o + t_{kj} \tag{6}$$

where:

$t_{nPV}$—time to fill a vehicle with a given capacity, depending on the rate of emptying it;

$t_m$—handling time related to the vehicle's arrival to the depot and preparatory activities for refueling (e.g., connecting the quick coupler) and back to the airport;

$t_o$—required standing time;

$t_{kj}$—time for the execution of fuel quality control procedure in the vehicle.

It was assumed that during $T_0$ flights, the vehicle could perform $N_{up}$ refueling cycles. The working time balance equation for one vehicle is needed to test the feasibility of the process under consideration, and can be written as follows:

$$t_{up} \cdot N_{up} + t_t \cdot N_t \leq T_0 \tag{7}$$

$$N_{up} \leq \frac{T_0}{t_{up}} \text{ and } N_t \leq \frac{T_0}{t_t} \tag{8}$$

where:

$t_{up}$—time of the vehicle refueling cycle;

$N_{up}$—number of refueling cycles of the vehicle;

$t_t$—aircraft refueling time *sp*;

$N_t$—number of refueled aircraft;

$T_0$—flight duration.

If $V_P \gg V_{zbsp}$, for a single vehicle it must be $N_{up} = N_t$, and there will be a time reserve; as a rule, $t_{up} \neq t_t$, and practically $t_t \ll t_{up}$.

The number of vehicles required by volume $N_{PV}$ for the transport of fuel consumed by the tactical fighter squadron should be converted into the number of vehicles physically needed, $N_P$, according to the formula:

$$N_P \geq \frac{N_{PV}}{N_{up}} \tag{9}$$

where:

$N_P$—number of vehicles supplying fuel to aircraft;
$N_{PV}$—number of vehicles required to carry the required amount of fuel by volume in one run;
$N_{up}$—number of possible refueling cycles for the vehicle.

Summing up, on the basis of the assumptions made, a set of the following equations/inequalities are obtained:

$$V_{eltmax} = \sum_{k}^{N_{SP}} \sum_{l}^{N_{elt}} \left( K_{zu}^{lk} \right) \cdot V_{zbsp} \tag{10}$$

$$N_{PV} \geq \frac{V_{eltmax}}{V_P} \tag{11}$$

$$N_P \geq \frac{N_{PV}}{N_{up}} \tag{12}$$

$$t_{up} \cdot N_{up} + t_t \cdot N_t \leq T_0 \tag{13}$$

$$V_P \cdot N_{up} \geq V_{zbsp} \cdot N_t \cdot K_{zu} \tag{14}$$

based on which it is possible to determine the set of solutions admissible within the adopted constraints. Next in the study, a numerical example is presented to illustrate the correctness and practical application of the calculation methodology described above.

## 3. Numerical Example

For the practical application of the developed calculation methodology, the following assumptions were made:

- Time of execution (duration) of flights performed by the tactical fighter squadron is $T_0 = 480$ (min);
- Number of aircraft participating in missions: 9;
- Flights are performed by Su-22 jets, where the capacity of the main fuel tank(s) of the aircraft is: $V_{zbsp} = 4625.0$ (dm$^3$);
- The capacity of the vehicle supplying fuel to the aircraft is: $V_P = 7500.0$ (dm$^3$);
- The fuel tank emptying factor of the aircraft depends on the distance of the flight and assumes the following values: $K_{zu} = \{0.33; 0.5; 0.66; 0.83\}$;
- Flights are executed in accordance with the provided flight schedule (Figure 3).

Taking into account the above assumptions, the minimum number of vehicles with a capacity of $V_P = 7500.0$ (dm$^3$), sufficient to secure an appropriate amount of aviation fuel, should be determined to ensure flight continuity.

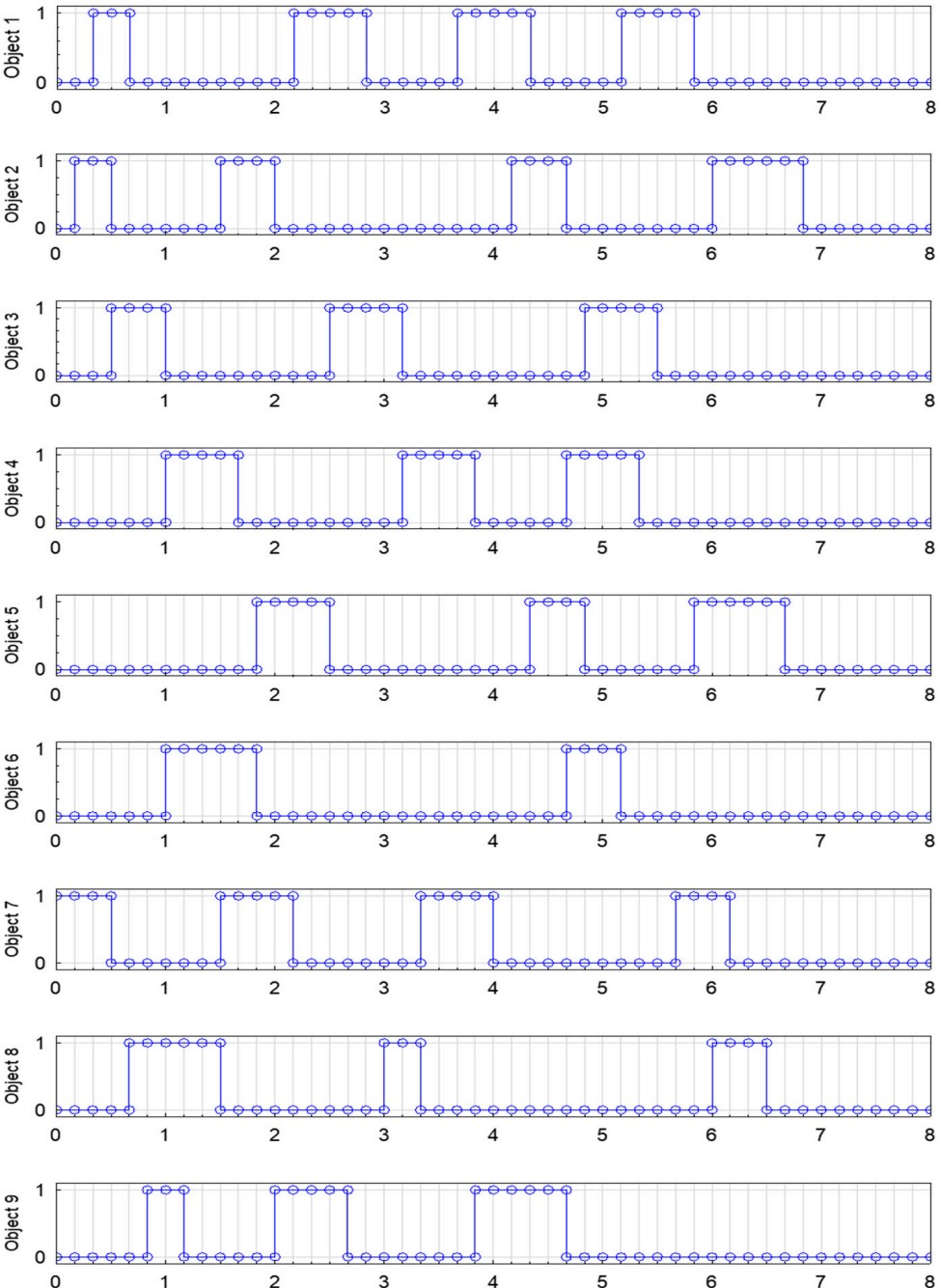

**Figure 3.** Planned flight schedule—variant. $T_0$—total flight time 480 (min); 0—waiting on the tarmac, ready for takeoff; 1—flight (performing combat tasks).

### 3.1. Solution

First of all, in accordance with Equation (10), the amount of fuel used by aircraft during flights should be calculated:

$$V_{eltmax} = \sum_{k}^{N_{SP}} \sum_{l}^{N_{elt}} \left( K_{zu}^{lk} \right) \cdot V_{zbsp} = 79{,}642.5 \ [\text{dm}^3] \tag{15}$$

The duration of the aircraft flight is described by a random step variable, depending on the type of tasks. Flight times are variables taking the following respective values: 20, 30, 40, and 50 min, which in the developed model is reflected by the coefficient $K_{zu}$ = {0.33; 0.5; 0.66; 0.83} of emptying the aircraft fuel tank.

### 3.1.1. Case Study I

Calculation of the amount of consumed aviation fuel for $K_{zu}$ = 0.83, for which the flight time is 50 min:

$$V_{elt0.83} = 0.83 \cdot \left( V_{2zbsp} + 2 \cdot V_{5zbsp} + V_{6zbsp} + V_{8zbsp} + V_{9zbsp} \right) = 23{,}032.5 \, [\text{dm}^3] \quad (16)$$

The specified number of vehicles with a fixed capacity in accordance with the relationship (11) must be greater than or equal to the required amount of fuel:

$$N_{PV} \cdot V_P \geq V_{elt0.83} \quad (17)$$

Therefore:

$$N_{PV} \geq \frac{V_{elt0.83}}{V_P} \geq 3.071 \approx 4 \quad (18)$$

The time of the vehicle refueling cycle ($t_{up}$) is the sum of the times needed for: vehicle arrival to the depot, handling, actual refueling time, time to return to the apron, and fuel settling time and checking its purity, which is, according to Formula (6):

$$t_{up} = 57.199 \, [\text{min}] \quad (19)$$

The aircraft refueling time ($t_t$) is strictly defined, as it depends on the $K_{zu}$ coefficient of aircraft tank emptying, the efficiency of the fuel distributor, and the time of the refueling vehicle reaching the aircraft. According to Equation (3), it was calculated as:

$$t_t = 17.79 \, [\text{min}] \quad (20)$$

The number of refueling cycles by both the vehicle $N_{up}$ and the number of aircraft refueling $N_t$ should be calculated in accordance with Equations (13) and (14) by solving the following formulas:

$$\begin{cases} t_{up} \cdot N_{up} + t_t \cdot N_t \leq T_0 \\ V_P \cdot N_{up} \geq V_{zbsp} \cdot N_t \cdot K_{zu} \end{cases} \quad (21)$$

$$N_{up} \leq \frac{T_0}{t_{up}} - \frac{t_t \cdot N_t}{t_{up}} \leq \frac{480}{57.199} - \frac{17.79 \cdot N_t}{57.199} \leq 8.39 - 0.311 N_t \quad (22)$$

$$N_{up} \geq \frac{V_{zbsp} \cdot N_t \cdot K_{zu}}{V_P} \geq 0.514 N_t \quad (23)$$

$$8.39 - 0.311 N_t = 0.514 N_t \quad (24)$$

$$0.825 N_t = 8.39 \Rightarrow N_t = 10.17 \quad (25)$$

$$N_{up} \geq 0.514 N_t \geq 5.22 \approx 6 \quad (26)$$

After substituting the calculated $N_{PV}$ and $N_{up}$ in Formula (12), we obtain:

$$N_{P0.83} \geq \frac{N_{PV}}{N_{up}} \geq 0.66 \quad (27)$$

### 3.1.2. Case Study II

Calculation of the amount of consumed aviation fuel for $K_{zu}$ = 0.66:

$$V_{elt0.66} = 0.66 \cdot \left( 3 \cdot V_{1zbsp} + 2 \cdot V_{3zbsp} + 3 \cdot V_{4zbsp} + V_{5zbsp} + 2 \cdot V_{7zbsp} + V_{9zbsp} \right) = 36{,}630 \, [\text{dm}^3] \quad (28)$$

The specified (minimum) number of vehicles with a fixed capacity in accordance with relationship (11) must be greater than or equal to the required amount of fuel, according to the following formulas:

$$N_{PV} \cdot V_P \geq V_{elt0.66} \tag{29}$$

$$N_{PV} \geq \frac{V_{elt0.66}}{V_P} \geq 4.88 \approx 5 \tag{30}$$

The time of the vehicle refueling cycle ($t_{up}$) is the sum of the times needed for: vehicle arrival to the depot, handling, actual refueling time, time to return to the apron, and fuel settling time and checking its purity, which for $K_{zu} = 0.66$ is:

$$t_{up} = 56.54 \, [\text{min}] \tag{31}$$

The aircraft refueling time ($t_t$) is strictly defined, as it depends on the $K_{zu}$ coefficient of aircraft tank emptying, the efficiency of the fuel distributor, and the time of the refueling vehicle reaching the aircraft:

$$t_t = 15.17 \, [\text{min}] \tag{32}$$

The number of refueling cycles by both the vehicle $N_{up}$ and the number of aircraft refueling $N_t$ should be calculated in accordance with Equations (13) and (14) by solving the following formulas:

$$t_{up} \cdot N_{up} + t_t \cdot N_t \leq T_0 \tag{33}$$

$$V_P \cdot N_{up} \geq V_{zbsp} \cdot N_t \cdot K_{zu} \tag{34}$$

$$N_{up} \leq \frac{T_0}{t_{up}} - \frac{t_t \cdot N_t}{t_{up}} \leq \frac{480}{56.54} - \frac{15.19 \cdot N_t}{56.54} \leq 8.48 - 0.268N_t \tag{35}$$

$$N_{up} \geq \frac{V_{zbsp} \cdot N_t \cdot K_{zu}}{V_P} \geq 0.407N_t \tag{36}$$

$$8.48 - 0.268N_t = 0.407N_t \tag{37}$$

$$0.675N_t = 8.48 \Rightarrow N_t = 12.56 \tag{38}$$

$$N_{up} \geq 0.407N_t \geq 5.11 \approx 6 \tag{39}$$

After substituting the calculated $N_{PV}$ and $N_{up}$ in Formula (12), we obtain the following condition:

$$N_{P0.66} \geq \frac{N_{PV}}{N_{up}} \geq 0.83 \tag{40}$$

### 3.1.3. Case Study III

Calculation of the amount of consumed aviation fuel for $K_{zu} = 0.5$ is:

$$V_{elt0.5} = 0.5\left(2 \cdot V_{2zbsp} + V_{3zbsp} + V_{5zbsp} + V_{6zbsp} + 2 \cdot V_{7zbsp} + V_{8zbsp}\right) = 16,187.5 \, [\text{dm}^3] \tag{41}$$

The specified number of vehicles with a fixed capacity must be greater than or equal to the required amount of fuel. It is therefore calculated according to the following formulas:

$$N_{PV} \cdot V_P \geq V_{elt0.5} \tag{42}$$

$$N_{PV} \geq \frac{V_{elt0.5}}{V_P} \geq 2.15 \approx 3 \tag{43}$$

The time $t_{up}$ of the vehicle refueling cycle, according to (6), is:

$$t_{up} = 55.927 \, [\text{min}] \tag{44}$$

The aircraft refueling time is calculated according to (3) as follows:

$$t_t = 12.71 \, [\text{min}] \tag{45}$$

The number of refueling cycles by both the vehicle $N_{up}$ and the number of aircraft refueling $N_t$ should be calculated in accordance with Equations (13) and (14) by solving the following set of formulas:

$$t_{up} \cdot N_{up} + t_t \cdot N_t \leq T_0 \tag{46}$$

$$V_P \cdot N_{up} \geq V_{zbsp} \cdot N_t \cdot K_{zu} \tag{47}$$

$$N_{up} \leq \frac{T_0}{t_{up}} - \frac{t_t \cdot N_t}{t_{up}} \leq \frac{480}{55.92} - \frac{12.71 \cdot N_t}{55.92} \leq 8.58 - 0.227 N_t \tag{48}$$

$$N_{up} \geq \frac{V_{zbsp} \cdot N_t \cdot K_{zu}}{V_P} \geq 0.308 N_t \tag{49}$$

$$8.58 - 0.227 N_t = 0.308 N_t \tag{50}$$

$$0.535 N_t = 8.58 \implies N_t = 16.03 \tag{51}$$

$$t_t = 12.71 \, [\text{min}] N_{up} \geq 0.308 N_t \geq 4.93 \approx 5 \tag{52}$$

After substituting the calculated $N_{PV}$ and $N_{up}$ in Formula (12), we obtain:

$$N_{P0.5} \geq \frac{N_{PV}}{N_{up}} \geq 0.6 \tag{53}$$

### 3.1.4. Case Study IV

Calculation of the amount of consumed aviation fuel for $K_{zu} = 0.33$ is:

$$V_{elt0.33} = 0.33 \left( V_{1zbsp} + V_{2zbsp} + V_{8zbsp} + V_{9zbsp} \right) = 6105.0 \, [\text{dm}^3] \tag{54}$$

The specified number of vehicles with a fixed capacity must be greater than or equal to the required amount of fuel. Thus, it is calculated as:

$$N_{PV} \cdot V_P \geq V_{elt0.33} \tag{55}$$

$$N_{PV} \geq \frac{V_{elt0.33}}{V_P} \geq 0.81 \approx 1 \tag{56}$$

The time $t_{up}$ of the vehicle refueling cycle is:

$$t_{up} = 55.27 \, [\text{min}] \tag{57}$$

The aircraft refueling time ($t_t$) is strictly defined, and is:

$$t_t = 10.05 \, [\text{min}] \tag{58}$$

The number of refueling cycles by both the vehicle $N_{up}$ and the number of aircraft refueling $N_t$ should be calculated in accordance with Equations (13) and (14) by solving the following set of formulas:

$$t_{up} \cdot N_{up} + t_t \cdot N_t \leq T_0 \tag{59}$$

$$V_P \cdot N_{up} \geq V_{zbsp} \cdot N_t \cdot K_{zu} \tag{60}$$

$$N_{up} \leq \frac{T_0}{t_{up}} - \frac{t_t \cdot N_t}{t_{up}} \leq \frac{480}{55.27} - \frac{10.05 \cdot N_t}{55.27} \leq 8.68 - 0.18 N_t \tag{61}$$

$$N_{up} \geq \frac{V_{zbsp} \cdot N_t \cdot K_{zu}}{V_P} \geq 0.203 N_t \tag{62}$$

$$8.68 - 0.18N_t = 0.203N_t \tag{63}$$

$$0.383N_t = 8.68 \Rightarrow N_t = 22.66 \tag{64}$$

$$N_{up} \geq 0.203N_t \geq 4.6 \approx 5 \tag{65}$$

After substituting the calculated $N_{PV}$ and $N_{up}$ in Formula (12), we obtain:

$$N_{P0.33} \geq \frac{N_{PV}}{N_{up}} \geq 0.2 \tag{66}$$

The required necessary number of vehicles with a capacity of 7500.0 (dm$^3$) was obtained as a result of summing up the partial values for the assumed $K_{zu}$ coefficients of the degree of emptying of the aircraft tank according to the relationship:

$$N_{P7.5} > N_{P0.33} + N_{P0.5} + N_{P0.66} + N_{P0.83} \Rightarrow N_{P7.5} > 2.29 \approx 3 \tag{67}$$

Thus, as it results from the numerical example, according to the assumed flight schedule (Figure 3), the necessary number of vehicles with a capacity of 7500.0 (dm$^3$) needed to secure the aircraft with aviation fuel is 3.

The proposed method is universal, and presents an example of the application of a mathematical apparatus enabling the combination of theory and practice in the field of modeling real processes implemented in the logistic system of an air base (BLot.).

## 4. Generalization of the Procedural Methodology and Summary of the Research Result

In an attempt to verify the correctness of the presented methodology of conduct and to emphasize its universality in this part of the study, calculations were made for all hypothetically possible scenarios of the execution of combat tasks for the tactical fighter squadron.

For this purpose, the following assumptions were made:

(1) The number of aircraft involved in flights is a random step variable within the range $X_{lsp} \in \{7, \dots, 16\}$, which in the model was reflected by the coefficient of efficiency $W_{efelt} \in \{0.43, \dots, 1\}$ of utilizing the squadron's aircraft;

(2) Aviation task completion times are $\{10; 20; 30; 40; 50\}$ minutes, respectively, which was represented in the model by the coefficient $K_{zu} = \{0.165; 0.33; 0.5; 0.66; 0.83\}$ of emptying the aircraft fuel tank;

(3) The maximum frequency of takeoffs is 40 min; in operational practice, this means the time necessary to perform the maintenance after each completed flight and to restore flight readiness (including refueling) before the next flight;

(4) According to the scheduled flight table, in the assumed time $T_0 = 480$ (min), each plane can perform at most a certain number of tasks depending both on its flight duration and frequency of takeoffs. Hence, a single aircraft during $T_0$ with $K_{zu}$ coefficients of emptying the aircraft fuel tank adopted (in point 2 above) can perform only a strictly defined number of aviation tasks; i.e.:

- For $K_{zu} = \{0.165\}$, the maximum number of possible aviation tasks during $T_0$ is 10;
- For $K_{zu} = \{0.33\}$, the maximum number of possible aviation tasks during $T_0$ is 8;
- For $K_{zu} = \{0.5\}$, the maximum number of possible aviation tasks during $T_0$ is 7;
- For $K_{zu} = \{0.66\}$, the maximum number of possible aviation tasks during $T_0$ is 6;
- For $K_{zu} = \{0.83\}$, the maximum number of possible aviation tasks during $T_0$ is 5.

Taking into account the above assumptions, 50 possible scenarios were considered, including a variable number of aircraft in the range of $7 \div 16$ and a variable flight time depending on the $K_{zu} = \{0.165; 0.33; 0.5; 0.66; 0.83\}$ of emptying the aircraft fuel tank, for which we calculated the necessary number of vehicles of a certain capacity needed to ensure reliable aviation fuel delivery to the aircraft performing combat flights.

The results of calculations for 50 possible scenarios related to two types of vehicles, i.e., with a capacity of 4500.0 (dm$^3$) and with a capacity of 7500.0 (dm$^3$) are presented in Figures 4 and 5, respectively.

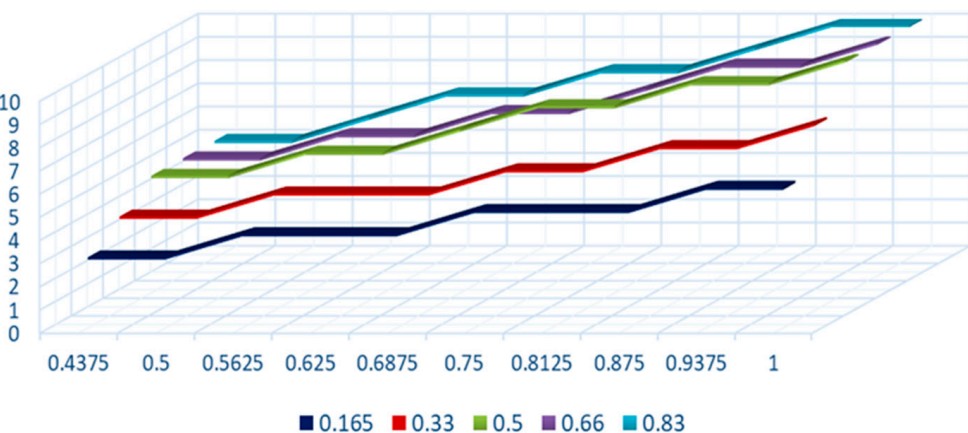

**Figure 4.** Calculated (real) number of vehicles with a capacity of 4500.0 (dm$^3$) necessary to secure aircraft fuel depending on the flight time and the efficiency coefficient $W_{efelt} \in \{0.43, \ldots 1\}$ of the utilization of the squadron's aircraft.

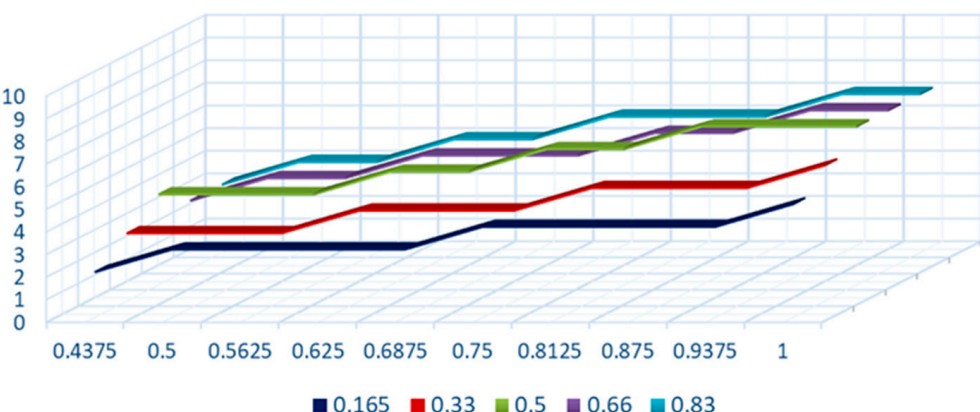

**Figure 5.** Calculated (real) number of vehicles with a capacity of 7500.0 (dm$^3$) necessary to secure aircraft fuel depending on the flight time and the efficiency coefficient $W_{efelt} \in \{0.43, \ldots 1\}$ of the utilization of the squadron's aircraft.

The curves shown in Figures 4 and 5 illustrate the minimum (necessary) number of vehicles of given capacities to meet the fuel needs of aircraft performing aviation tasks. They reflect the utilization rate of tactical fighter squadron's airplanes through the coefficient $W_{efelt} \in \{0.43, \ldots, 1\}$ corresponding to the number of aircraft from 7 to 16. Efficiency of using 100% of the assets (i.e., all 16 aircraft) is important from an operational point of view, as it translates into a collective indicator of technical readiness of the tactical fighter squadron.

Summing up the results obtained for all 50 considered scenarios of the planned flight schedule (Figure 3) and all flight times and frequencies of takeoffs, we concluded that:

- The minimum number of vehicles with the capacity of 7500.0 (dm$^3$) necessary to ensure the continuity of supply of aircraft fuel will range from 2 to 7 (Figure 5);
- The minimum number of vehicles with the capacity of 4500.0 (dm$^3$) necessary to ensure the continuity of supply of aircraft fuel will range from 3 to 10 (Figure 4).

In an attempt to generalize the proposed methodology and determine the set of the most probable solutions, calculations were performed for all possible cases (i.e., 50 scenarios). Their results are summarized in Tables 3 and 4.

**Table 3.** The minimum number of vehicles with a capacity of 7500.0 (dm³) necessary to secure aircraft fuel depending on the flight time and the efficiency coefficient $W_{efelt} \in \{0.43, \ldots, 1\}$ of utilization of the squadron's airplanes.

| $W_{efelt}$ | 0.4375 | 0.5 | 0.5625 | 0.625 | 0.6875 | 0.75 | 0.8125 | 0.875 | 0.9375 | 1 |
|---|---|---|---|---|---|---|---|---|---|---|
| $K_{zu} = 0.165$ | 2 | 3 | 3 | 3 | 3 | 4 | 4 | 4 | 4 * | 5 |
| $K_{zu} = 0.33$ | 3 | 3 | 3 | 4 | 4 | 4 | 5 | 5 | 5 | 6 |
| $K_{zu} = 0.5$ | 4 | 4 | 4 | 5 | 5 | 6 | 6 | 7 | 7 | 7 |
| $K_{zu} = 0.66$ | 3 | 4 | 4 | 5 | 5 | 5 | 6 | 6 | 7 | 7 |
| $K_{zu} = 0.83$ | 3 | 4 | 4 | 5 | 5 | 6 | 6 | 6 | 7 | 7 |

* the most probable solution.

**Table 4.** The minimum number of vehicles with a capacity of 4500.0 (dm³) necessary to secure aircraft fuel depending on the flight time and the efficiency coefficient $W_{efelt} \in \{0.43, \ldots, 1\}$ of utilization of the squadron's airplanes.

| $W_{efelt}$ | 0.4375 | 0.5 | 0.5625 | 0.625 | 0.6875 | 0.75 | 0.8125 | 0.875 | 0.9375 | 1 |
|---|---|---|---|---|---|---|---|---|---|---|
| $K_{zu} = 0.165$ | 3 | 3 | 4 | 4 | 4 | 5 | 5 | 5 * | 6 | 6 |
| $K_{zu} = 0.33$ | 4 | 4 | 5 | 5 | 5 | 6 | 6 | 7 | 7 | 8 |
| $K_{zu} = 0.5$ | 5 | 5 | 6 | 6 | 7 | 8 | 8 | 9 | 9 | 10 |
| $K_{zu} = 0.66$ | 5 | 5 | 6 | 6 | 7 | 7 | 8 | 9 | 9 | 10 |
| $K_{zu} = 0.83$ | 5 | 5 | 6 | 7 | 7 | 8 | 8 | 9 | 10 | 10 |

* the most probable solution.

Figure 6 presents solutions that collectively present the number of necessary vehicles of a certain capacity from a set of 50 adopted scenarios.

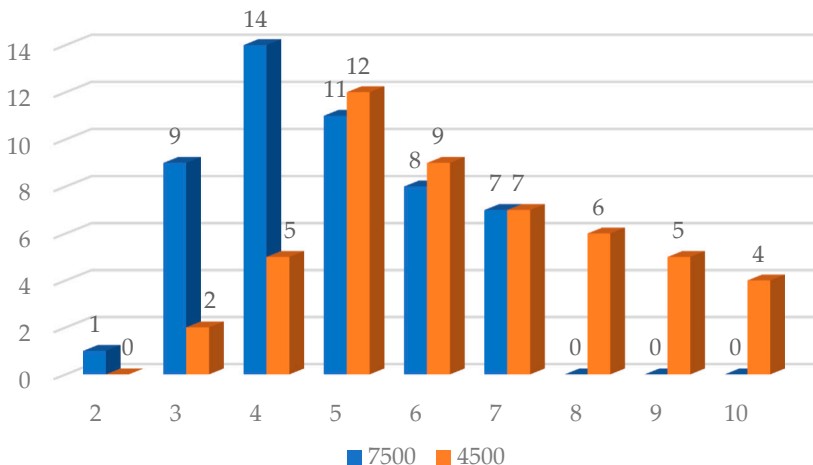

**Figure 6.** The most probable solutions (dominants) in the set of 50 possible scenarios for vehicles with a capacity of 7500.0 (dm³) and 4500.0 (dm³).

When analyzing the set of acceptable solutions (Figure 6) for both types of vehicles, it should be stated that:

- In the case of a 4500.0 (dm³) capacity vehicle, the dominant were four vehicles in the set of possible solutions with a frequency of 0.28% (14/50);
- In the case of a 7500.0 (dm³) capacity vehicle, the dominant were five vehicles in the set of possible solutions with a frequency of 0.24% (12/50).

Table 5 summarizes the total fuel consumption in the set of all possible scenarios, which is proportional to the length of a single flight and the efficiency coefficient of the squadron's aircraft use.

**Table 5.** Total fuel consumption (dm$^3$) of the tactical fighter squadron during combat flights depending on the length of a single flight and the efficiency coefficient $W_{efelt} \in \{0.43, \ldots, 1\}$ of the squadron's aircraft use.

| $W_{efelt}$ | 0.4375 | 0.5 | 0.5625 | 0.625 | 0.6875 | 0.75 | 0.8125 | 0.875 | 0.9375 | 1 |
|---|---|---|---|---|---|---|---|---|---|---|
| $K_{zu} = 0.165$ | 54,319 | 61,050 | 68,681 | 76,313 | 83,944 | 91,575 | 99,206 | 106,838 | 114,469 | 122,100 |
| $K_{zu} = 0.33$ | 85,470 | 97,680 | 109,890 | 122,100 | 134,310 | 146,520 | 158,730 | 170,940 | 183,150 | 195,360 |
| $K_{zu} = 0.50$ | 113,313 | 129,500 | 145,688 | 161,875 | 178,063 | 194,250 | 210,438 | 226,625 | 242,813 | 259,000 |
| $K_{zu} = 0.66$ | 128,205 | 146,520 | 164,835 | 183,150 | 201,465 | 219,780 | 238,095 | 256,410 | 274,725 | 293,040 |
| $K_{zu} = 0.83$ | 134,356 | 153,550 | 172,744 | 191,938 | 211,131 | 230,325 | 249,519 | 268,713 | 287,906 | 307,100 |

This is characterized by significant variability; its disproportion in the range of max/min values is nearly 6 times (=5.749), and is greater for the longest flights $K_{zu} = \{0.83\}$ compared to the shortest ones $K_{zu} = \{0.165\}$, taking into account the efficiency coefficient in the range $\{0.43, \ldots, 1\}$. It directly translates into the number of vehicles needed by volume (i.e., in one trip) to carry the necessary amount of fuel, where for vehicles with a capacity of 7500.0 (dm$^3$), this number ranges from 8–41 (Table 1); and for vehicles with a capacity of 4500.0 (dm$^3$), it ranges from 12–69 (Table 2). However, neither the number of vehicles required by volume to carry fuel in one trip nor the total fuel consumption itself have a decisive influence on the minimum number of vehicles necessary to ensure the supply of aviation fuel during flights performed by the squadron. This is confirmed by the detailed results summarized in Tables 1–4, as well as the dominant graph (Figure 6) for both types of vehicles. For a vehicle with a capacity of 7500.0 (dm$^3$) in the set of permissible solutions, the dominant was four vehicles, with the probability of its observation amounting to 0.28 (14/50). The disproportion of fuel used in the min/max range was almost 1:2 (1:1.89—see Table 5), and was almost proportional to the number of vehicles required by volume to transport all fuel in one trip, variable in the range of 13–24 vehicles (compare the data presented in the Tables 1 and 3). Thus, nearly doubling the amount of fuel required did not have a decisive impact on the minimum number of vehicles of four (see blue cells in Table 3). Similar conclusions can be drawn for vehicles with a capacity of 4500.0 (dm$^3$) in the set of permissible solutions, the dominant was five vehicles (Figure 6), and the probability of its occurrence was 0.24 (12/50). In this case, the disproportion of the used fuel in the min/max range was almost 1:1.8, which translated into the number of vehicles required by volume to transport all fuel in one trip varies in the range of 17–30 (compare the data summarized in Tables 2 and 4). In addition, in this case, the almost two-fold difference in the amount of fuel used did not have a decisive impact on the minimum number of vehicles amounting to five (see cells marked in red in Table 4). The minimum number of vehicles necessary to provide fuel is additionally influenced by other factors, which include: the assumed maximum frequency of flights, the time $t_{up}$ of the refueling cycle of each vehicle, time $t_t$ of refueling the aircraft, and the proportion/quotient of aircraft refueled before the vehicle has to drive to the depot to refill. They are listed in Table 6.

**Table 6.** Summary of the aircraft refueling time $t_t$, the time of the refueling cycle by the vehicle $t_{up}$, and the proportion $t_t/t_{up}$ depending on the $K_{zu}$ factor of emptying the aircraft tank.

| | $K_{zu} = 0.165$ | $K_{zu} = 0.33$ | $K_{zu} = 0.5$ | $K_{zu} = 0.66$ | $K_{zu} = 0.83$ |
|---|---|---|---|---|---|
| $t_t$ (min) | 7.54 | 10.09 | 12.71 | 15.18 | 17.80 |
| $t_{up}$ (min) | 54.63 | 55.27 | 55.93 | 56.54 | 57.19 |
| $t_t/t_{up}$ | 0.138073 | 0.182507 | 0.227230 | 0.268376 | 0.3111216 |

The times of refueling the aircraft $t_t$, the cycle of refueling by the vehicle $t_{up}$, and the quotient $t_t/t_{up}$ presented in Table 6 are all directly related to the flight duration of a single aircraft. The refueling times $t_t$ and the refilling cycle of the vehicle $t_{up}$ are constant and depend on the $K_{zu}$ factor of the aircraft's fuel tank emptying. The $t_t/t_{up}$ quotient is in the range $0.138 \div 0.311$ (Table 6) and increases with the increase in the duration of a single flight. Based on the data compiled in Tables 1–4, the following regularity can be formulated; i.e., the longer the time of a single flight and the greater the disproportion $t_t/t_{up}$, the greater the minimum number of vehicles is necessary to provide enough fuel to the aircraft during flights. Additionally, taking into account that the capacity of both vehicles 4500.0 $(dm^3)$ and 7500.0 $(dm^3)$ are close to the maximum capacity of the aircraft tank (for the Su-22, it is 4625.0 $(dm^3)$), the emptying factors for flight times of 40 (min) and 50 (min), for which $K_{zu}$ amounted to 0.66 and 0.83, respectively, the vehicle had to complete the refueling cycle after each refueling of the aircraft.

## 5. Conclusions

The purpose of this article was to develop a method that would enable us to determine the minimum number of vehicles supplying aviation fuel to aircraft during combat tasks. Tactical fighter squadrons, equipped with 16 aircraft in the Polish Armed Forces, carry out combat tasks in accordance with the planned flight schedule (Figure 3). In operational practice, flights are carried out only by operational aircraft, the number of which depends on the adopted repair and maintenance planning strategy, and the type of combat tasks to be performed. When it comes to tactical fighter squadrons, combat tasks are usually performed using between 7 and 10 aircraft.

In operational practice, the number of refueling vehicles needed to secure jet fuel is determined arbitrarily, based on experience with a certain excess, in order to ensure the reliability of the supply system. Taking into account the above-mentioned arguments, the paper proposed a proprietary calculation methodology enabling the determination of the minimum number of vehicles necessary to supply aircraft with jet fuel.

The method was verified with a numerical example confirming its correctness. For its development, systems of equations/formulas were used, on the basis of which the minimum number of vehicles supplying aircraft with fuel was determined. It should be remembered that the method depended on a series of factors (input data), which included:

- Type and number of aircraft performing combat tasks (in this study, these were Su-22 aircraft);
- Planned flight schedule specifying for each aircraft individually: the duration (time) of the flight, as well as the frequency of each takeoff;
- Type of vehicle supplying aviation fuel (for the Su-22, these are vehicles with a capacity of 4500.0 $(dm^3)$ and 7500.0 $(dm^3)$);
- Organization of the system of vehicle refilling, taking into account handling times (travel time to the fuel depot, time of the vehicle refilling process, while taking into account the efficiency of the dispenser and the time required to return to the apron), fuel settling time, quality control, and crew training level.

In the example considered in this study, the flight time was eight hours, and nine Su-22 airplanes were used. Vehicles delivering fuel to the aircraft during the flights performed had an assumed capacity of 7500.0 $(dm^3)$. According to the obtained computation result, the minimum number of these vehicles necessary to ensure the continuity of aviation fuel supply was three (see Section 3). The drawback of the method was that the calculations were performed in stages; i.e., depending on the flight distance reflected by the $K_{zu}$ factor of the fuel consumption of the aircraft. The developed model can be modified by introducing an excess factor, which will increase the necessary number of vehicles and thus increase the reliability of the aircraft fuel supply system.

**Author Contributions:** Conceptualization, J.Z. and J.Ż.; methodology, J.Z.; software, J.Z.; validation, J.Z., M.O. and J.S.-R.; formal analysis, J.Z. and J.Ż.; investigation, J.Z. and J.Ż.; resources, J.Z.;

data curation, J.Z. and J.M.; writing—original draft preparation, J.Z. and J.S.-R.; writing—review and editing, J.Z., J.M. and M.O.; visualization, J.Z. and M.O.; supervision, J.Z. and M.O.; project administration, J.Z. and J.S.-R.; funding acquisition, J.Z. and J.M. All authors have read and agreed to the published version of the manuscript.

**Funding:** This work was supported by the Military University of Technology (project No. 878/WAT/2021). This support is gratefully acknowledged.

**Conflicts of Interest:** The authors declare no conflict of interest.

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
