# Peer review of "Method for Calculating the Required Number of Transport Vehicles Supplying Aviation Fuel to Aircraft during Combat Tasks"

_sustainability, doi:10.3390/su14031619_

Round 1

Reviewer 1 Report

The manuscript deals with the minimum number of vehicles supplying fuel to aircrafts during combat tasks. In the reviewer's opinion, the manuscript should be significantly improved. The following issues can be considered:

1. How many aircrafts are considered? 16 or 9?

2. The proposed method subjected to many assumptions, is the method appliable to real situation?

3. How to verify the present method?

4. In spite of well-presnted theoretical background, the results seem to be insufficiant.

5. There are many equations cannot be correctly displayed in the pdf file.

Author Response

Dear Reviewer #1,

We would like to sincerely thank you for the effort in a very thorough analysis of the article in terms of its scientific aspects. We would also thank for all constructive comments and suggestions as well as for seeing potential in our work. We have revised the whole manuscript according to your constructive suggestions, which has greatly improved the quality and the presentation of the paper. Below, please find our responses to all your remarks/doubts.

Additionally, we would like to inform the reviewers, that the final version of the article was proofread by a professional translation agency.

According to the submitted questions/doubts:

The manuscript deals with the minimum number of vehicles supplying fuel to aircrafts during combat tasks. In the reviewer's opinion, the manuscript should be significantly improved. The following issues can be considered:

  1. How many aircrafts are considered? 16 or 9?

In the example under consideration (see point 3. Numerical example) there was presented an actual planned flight schedule from the air base operating within the 2nd Tactical Air Wing of the Polish Armed Forces, according to which combat tasks were performed by 9 aircraft (see Figure 3. Planned flight schedule - variant). Additionally, the number of aircraft was recorded (see point 3, line 252 "number of aircraft participating in missions: 9"). On the other hand, the developed method makes it possible to determine the necessary number of aerial refuelling vehicles for any number of aircraft (specifically Su-22), whose maximum number in a squadron is 16 objects.

  1. The proposed method subjected to many assumptions, is the method appliable to real situation?

All assumptions presented in this paper are based on real operational data (aircraft type, tank vehicle type, individual flight lengths and departure frequencies, calculated calculation-manipulation times, etc.). The added value of the developed method is the fact that it is dedicated to real combat operations where, due to the losses incurred, there is a significant probability (e.g. due to damage) of a deficit of military equipment (SpW) in the form of vehicles providing aviation fuel (time "W" of war).

  1. How to verify the present method?

The new calculation methodology has been verified by means of an example (see point 3. Numerical example) showing the sequence of calculations performed and, as a result, how it is applied. Then, detailed calculations (see point 4. Generalization of the procedural methodology and summary of the research result) were performed for potentially available 50 scenarios in order to verify the reliability of the developed model.

  1. In spite of well-presented theoretical background, the results seem to be insufficient.

Theoretical fundamentals of the method and assumptions concerning the mathematical model are presented in section 2 (see 2. Assumptions and development of the mathematical model). The adopted limitations result from the real conditions of combat flights implementation with the use of military aircraft of Su-22 type. The developed theoretical basis in the form of assumptions, adopted designations and written equations and inequalities de facto create an added value, because they reflect the developed mathematical model used to determine the necessary number of vehicles supplying aircraft with aviation fuel. Verification of the correctness of the developed method was made in two stages, i.e. by presenting in point 3 (see point 3. Numerical example) the method of performing calculations, in which for the assumed real schedule (table) of flights, the necessary (minimum) number of such vehicles with a capacity of 7500 [dm3] was calculated and in point 4 (see point 4. Generalization of the procedural methodology and summary of the research result), in which a generalization of the method and its universality was proposed. In the mentioned part of the article firstly: the results of calculations for potentially possible 50 scenarios (planned flight tables) are presented, secondly: the calculations were carried out for two types of vehicles with capacities respectively: 7500 [dm3] (see Table 3) and 4500 [dm3] (see Table 4). Third: for each solution dominants were calculated (Figure 6. The most probable solutions (dominants) in the set of 50 possible scenarios for vehicles with a capacity of 7,500 [dm3] and 4,500 [dm3]), and then probabilities of their occurrence in the set of acceptable solutions were determined.

  1. There are many equations cannot be correctly displayed in the pdf file.

The presented illegible characters in the equations are most likely the result of converting the authors' edited version of the file to a PDF file by the software tool used by the publisher. The authors guarantee that in the original and editable version all equations are correctly written and will pay attention to this problem when preparing the final version of the article for publication.

Reviewer 2 Report

Dear authors

The corrections comments are based on the manuscript you were followed.

Concern # 1: in the abstract, include the final result of the paper.

Concern # 2: in the introduction, clarify the contribution of your paper.

Concern # 3: Tables 1 and 2, need more discussion.

Concern # 4: some strange symbols in equations 7, 21, 22,23, 25, 29, 33, 34,35,36,38,42, 46, 47, 48, 49, 55, 59, 60, 61, and 62. Correct the above equations.

Concern # 5: Many references are not related to the paper subject, please revise the references.

Author Response

Dear Reviewer #2,

We would like to sincerely thank you for the effort in a very thorough analysis of the article in terms of its scientific aspects. We would also thank for all constructive comments and suggestions as well as for seeing potential in our work. We have revised the whole manuscript according to your constructive suggestions, which has greatly improved the quality and the presentation of the paper. Below, please find our responses to all your remarks/doubts.

Additionally, we would like to inform the reviewers, that the final version of the article was proofread by a professional translation agency.

According to the submitted questions/doubts:

Dear authors

The corrections comments are based on the manuscript you were followed.

Concern # 1: in the abstract, include the final result of the paper.

This paper develops a method for determining the minimum number of vehicles required to supply aircraft (sp) with aviation fuels. The developed method was verified by a numerical example illustrating its application in practice. In addition, a detailed analysis of its application was carried out in relation to potentially possible 50 scenarios of combat task execution, with a number of assumptions fulfilled. Based on the performed calculations, it can be concluded that the number of vehicles required for sp fuel supply depends on several factors: the number of aircraft, the characteristics of air tasks (flight length and frequency of departures), as well as the time of clean sp refuelling and the duration of the vehicle-tanker refuelling cycle. (see Abstract, lines 16-24)

Concern # 2: in the introduction, clarify the contribution of your paper.

In this study, the authors dealt with the original and unique issue related to the problem of determining the minimum number of vehicles supplying aviation fuel to aircraft performing combat tasks. The Su-22 combat fighter aircraft (manufactured in the USSR) is still used by the air force units of the Republic of Poland (RP) (see Chapter 1. Introduction, lines 82-84). During the execution of combat tasks, after departure, the readiness for the next flight is regained each time, the essential element of which is refueling. The refueling process is carried out using two types of tanker vehicles of different tank capacities, 4,500.0 [dm3] and 7,500.0 [dm3], respectively. According to the operational practice, the number of vehicles of a certain type needed to secure aviation fuel is each time determined arbitrarily with a set operational (equipment) surplus. This is due to safety reasons and is dictated by the reliability of the fuel supply system ensuring the performance of planned combat tasks. The number of fuel delivery vehicles depends on a number of factors, including the number and type of air-craft (sp) involved in the flights, the capacity of the tank(s) of the main (or all) aircraft and its emptying rate. The adopted organization of flights is also important, including both the length of individual flights and the frequency of departures individually for each aircraft. The proposed method in this publication enables the determination of the necessary (minimum) number of vehicles needed to secure necessary aviation fuel for the aircraft. It is dedicated especially to aviation tasks carried out in emergency situations, for example during war operations, when random losses of military equipment (ME), both in terms of primary and supporting equipment often being of essential operational importance are characteristic. The developed method is universal and can be applied to any type/kind of aircraft and any combat flight scenario.

In order to present the solution to this original problem, in which the authors have offered their contribution to the solution of a unique issue, the following publication layout has been proposed. Chapter 1 presents a literature review of linear programming, which is a method used to achieve the best (optimal) solution in the decision-making process. In chapter 2 all assumptions and development of a mathematical model developed on the basis of an analysis of the real process of aviation fuel supply of Su-22 aircraft performing combat tasks have been presented. The proposed model is an original scientific achievement of the authors of this publication. Chapter 3 illustrates the application of the developed model on a numerical example related to the real scheduled flight table (real case) reflecting the combat tasks performed by the tactical aviation base of the Polish Armed Forces. Chapter 4 illustrates the extension and generalization of the developed mathematical model by carrying out calculations for 50 potential (possible) action scenarios. Obtained results are a confirmation of correctness and at the same time universality of the proposed model. Finally, in chapter 5 the obtained solutions indicating shortcomings and factors affecting the obtained results have been discussed and concluded. (see Chapter 1. Introduction, lines 104-119).

Concern # 3: Tables 1 and 2, need more discussion.

The results summarized in Table 1 and Table 2 show the ranges of variation in the number of vehicles with specific capacities, respectively (7500.0 [dm3] for Table 1 and 4500.0 [dm3] for Table 2), needed by volume to deliver the necessary amount of fuel (per flight) to secure flights. The Wefelt coefficient of aircraft utilization efficiency ranging from 43.75% to 100% means that 7 to 16 aircraft on the equipment of the tactical aviation squadron are actively involved in the execution of combat tasks. As can be seen from the above tables, the needs for fuel supply directly translate into the number of vehicles of a certain capacity according to the principle of the smaller the capacity, the more vehicles should be secured to carry the required amount of fuel in one trip. (see Chapter 2. Assumptions and development of the mathematical model, lines 198-206)

Concern # 4: some strange symbols in equations 7, 21, 22,23, 25, 29, 33, 34,35,36,38,42, 46, 47, 48, 49, 55, 59, 60, 61, and 62. Correct the above equations.

The presented illegible characters in the equations are most likely the result of converting the authors' edited version of the file to a PDF file by the software tool used by the publisher. The authors guarantee that in the original and editable version all equations are correctly written and will pay attention to this problem when preparing the final version of the article for publication.

Concern # 5: Many references are not related to the paper subject, please revise the references.

The paper presents a new method for determining the necessary number of vehicles supplying aviation fuel to aircraft performing combat tasks. In operational practice, such a number is determined arbitrarily (experimentally) with a certain excess due to the existing risk of not performing the assumed actions caused by a lack of fuel supply. The mathematical model developed in Chapter 2 is related to linear programming problems, the main area of which are optimization problems. For this reason, the literature review performed in Chapter 1. Introduction  deals precisely with the application of linear programming with particular emphasis on optimization problems. With the above in mind, the authors extended the literature review with additional 3 items, indicated below, which substantively underpin the theoretical part of this paper.

[11]      Janekova J, Fabianova J, Kadarova J. Selection of optimal investment variant based on monte carlo simulations. Int J Simul Model 2021;20:279–90. https://doi.org/10.2507/IJSIMM20-2-557.

[16]      Huang C, Cheng X. Estimation of aircraft fuel consumption by modelling flight data from avionics systems. J Air Transp Manag 2022;99. https://doi.org/10.1016/j.jairtraman.2022.102181.

[32]      GoÅ‚da P, Zawisza T, Izdebski M. Evaluation of efficiency and reliability of airport processes using simulation tools. Eksploat Niezawodn 2021;23:659–69. https://doi.org/10.17531/ein.2021.4.8.

Round 2

Reviewer 1 Report

The authors have improved the manuscript and addressed the questions quite well. This version is well presented and satisfactory. Therefore, it is recommended to accept in current form.